

# On measuring snow ablation rates in alpine terrain with a mobile GPR device

Nena Griessinger[1,2], Franziska Mohr[2], and Tobias Jonas[1]

[1]WSL Institute for Snow and Avalanche Research SLF, Davos, Switzerland
[2]Department of Geography, University of Zurich, Zurich, Switzerland

*Correspondence to:* Nena Griessinger (nena.griessinger@slf.ch)

**Abstract.** Ground penetrating radar (GPR) has become a promising technique in the field of snow hydrological research. It is commonly used to measure snow depth, density, and water equivalent over large distances or along gridded snow courses. Having built and tested a mobile light-weight setup, we demonstrate that GPR is capable of accurately measuring snow ablation rates in complex alpine terrain. Our setup was optimized for efficient measurements and consisted of a common-mid-point assembly with four pairs of antennas mounted to a plastic sled, which was small enough to permit safe and convenient operations. Repeated measurements were taken during the 2014/15 winter season along ten profiles within two valleys located in the eastern Swiss Alps. Resulting GPR-based data of snow depth and water equivalent as well as their respective change rates over time were in good agreement with concurrent manual measurements, in particular if accurate alignment between repeated overpasses could be achieved (root-mean-square error of 4.5 cm for snow depth, 25 mm for snow water equivalent, and 4.4 cm and 26 mm for the respective change rates). With its suitability for alpine terrain and the achieved accuracy, the presented setup could become a valuable tool to validate snowmelt models or to complement lidar-based snow surveys.

## 1  Introduction

Terrestrial and airborne laser scanning of snow has significantly increased our ability to understand the spatial variability of snow depth (Harper and Bradford, 2003; Deems et al., 2013). However, methods to provide corresponding measurements of snow water equivalent of similar range and quality, particularly during the melt phase, are yet unavailable. Ground penetrating radar (GPR) technology allows estimating snow properties such as depth, density, and water equivalent. But most applications that cover large areas do currently not have sufficient accuracy to study spatial variability of snow density in detail and still depend on concurrent ground-based calibration data (Lundberg et al., 2010). Nevertheless, ground penetrating radar (GPR) has become more accessible to snow researchers and is already successfully being used in the context of snow hydrological studies for both calibration and validation purposes. GPR presents a non-destructive method and is, compared to manual measurements, very fast, which constitutes the potential of this technology for extensive surveys of snow properties.

GPR has become a frequently applied technique in glaciology (Plewes and Hubbard, 2001; Woodward and Burke, 2007; Forte et al., 2013; Booth et al., 2013) and snow research (Bradford et al., 2006; Sundström et al., 2013; Gustafsson et al., 2012; Lundberg et al., 2006; Lundberg and Thunehed, 2000; Lundberg et al., 2000; Marchand, 2003; Marshall and Koh, 2008; Ulriksen,



1982; Yamamoto et al., 2004). Lundberg et al. (2010) give a valuable review on the use of GPR in snow studies carried out in Norway, Finland, and Sweden. GPR measurements in Scandinavia, for example, are routinely carried out from snowmobiles to estimate catchment-scale snow water resources (Andersen et al., 1987; Marchand et al., 2001; Marchand and Killingtveit, 2001, 2004).

5  Many applications of GPR to measure spatially distributed snow properties are generally conducted when dry snow conditions are present. Under these conditions, measurements of snow depth, snow water equivalent and snow density are comparably easy to obtain and can be quite accurate (Andersen et al., 1987; Sand and Bruland, 1998). Sand and Bruland (1998), as well as others, have assumed snow density to be approximately constant along a measured profile when estimating snow water equivalent from the GPR data. Alternatively, Lundberg et al. (2006) assumed density to be linearly dependent on snow depth.

10  However, when liquid water is present in the snowpack, data analysis methods become more complex and snow water equivalent estimations typically more uncertain (Lundberg and Thunehed, 2000). Bradford et al. (2006) determined the liquid water content by using the frequency shift method to estimate the complex electrical permittivity and by using the common-mid-point (CMP) method to estimate the real part of the electrical permittivity. Another approach to determine the liquid water content, proposed by Sundström et al. (2013), is based on the effective electrical conductivity obtained from the two-way-travel time, the propagation velocity, and the attenuation of a radar wave within a snowpack.

15  GPR technology has been used in many ways to measure snow properties. Here we focus on approaches that allow for spatially distributed measurements, in particular along line or gridded transects. To measure snowpack properties in the presence of density changes, Gustafsson et al. (2012) used an array of multiple impulse antennas of two different nominal frequencies in a row mounted on a sled, which was pulled by a snowmobile, and tested it along a 1 km line. We adopted this approach to develop a light-weight mobile GPR system that allows simultaneous estimation of snow depth, snow density, and snow water equivalent in a snow survey setting in alpine terrain. For this purpose, we built a GPR platform on a sled system with four antenna pairs set up as a CMP array. We will show that this approach represents a very efficient means of spatially-distributed measurements that are similarly accurate as manual measurements.

This study elaborates further on the question whether GPR systems of this type can possibly be used to determine spatially distributed snow ablation rates in alpine terrain. To this end, we used the GPR system to retrieve snow properties along several preselected transects. Careful repetition of these measurements along the same transects after several days of snowmelt provided both absolute values as well as corresponding change rates, which we denote as snow ablation rates. Manual snow samples of snow depth and snow water equivalent were collected during the field campaigns for validation purposes. These data allow demonstrating the feasibility of using mobile GPR systems to derive snowmelt rates and examining the practicalities of such measurements.





## 2 Methods

### 2.1 Study areas

Measurements were conducted in two valleys in the vicinity of Davos, Switzerland, during the winter 2014/15. Field sites were selected according to different requirements. The set of sites should cover a certain range of elevations, aspects, and slope

inclinations. Further, each individual site needed to be clear of recreational as well as dangerous areas, in particular avalanche runout zones. Avoiding avalanche exposed terrain was not only for safety reasons but also because an avalanche would have interrupted the series of repeated measurements. Seven sites were located in the Monbiel valley (Fig. 1 top right) between 1300 and 1400 m a.s.l. where snowmelt occurred from mid-March 2015 onwards. Starting mid-April 2015, measurements were taken in the Sertig valley (Fig. 1 bottom right) between 1850 and 1950 m a.s.l. at three sites. In both valleys, our sites were mostly

located on snow-covered alpine meadows. Table 1 details on the characteristics of all transects.

### 2.2 Measurements

A variety of procedures were applied during fieldwork at each transect. The sled was towed by two persons, one to the left and one to the right, so that it was not required to step on the transects. That way, the GPR assembly was moved at an average speed of approximately $0.4\,\mathrm{m\,s^{-1}}$. For reference, we periodically positioned markers (bamboo sticks) along the transects. These

reference points were marked in the radargrams to allow exact synchronization between repeated measurements along the same transect. Manual measurements of snow depth, snow water equivalent, and liquid water content were taken along each transect at the reference points and used for calibration and validation purposes. Subsequent overpasses followed the tracks of the sled drawn during the first acquisition and allowed achieving accurate spatial match between multiple overpasses of the same transect. To avoid the sled from breaking through the snowpack and destroying the tracks, the fieldwork was carried out

when a crust had formed after cold, clear nights. Measurements at each transect could be repeated three to five times depending on the remaining snow height, as well as meteorological and logistical conditions.

### 2.2.1 Construction and setup of the mobile GPR system

Typically, snowpack properties such as snow depth, snow water equivalent and snow density show strong lateral variations. To measure those simultaneously, the use of a multi-offset approach is indicated. To allow for efficient sampling we opted for a

fixed assembly mounted to a plastic sled. At the same time, the assembly needed to be light and small enough to permit safe and convenient operations in snow-covered rugged alpine terrain. Weight could be saved considerably by adopting an antenna setup suggested by Gustafsson et al. (2012) and referred to as a shifted CMP design. This design is illustrated in Fig. 2 and consists of four instead of eight antennas to form four parings, which is achieved by combining each of the transmitting with each of the receiver antennas. The downside of this approach is that each pairing has a different midpoint which needs to be

synchronized in the post processing. While this is neither difficult nor computationally intensive, it does require sampling at fixed spatial intervals (as opposed to fixed temporal intervals), which necessitates an accurate spatial referencing system.



We used a MALA GPR ProEx (MALA Geoscience, Malå Sweden) system with four of their separable shielded antennas, featuring a nominal frequency of 1300 MHz. The CMP array was mounted on a light-weight plastic sled (HDPE pulk, Snowsled, UK) with a large baseboard to form a level bottom surface, which provided the fixed antenna positions during the measurements (Fig. 2). This way, the antennas were placed approximately 2 cm above the snow surface. To keep the system free from

snow, all GPR components including the main unit, antennas, batteries, and cables were enveloped in a waterproofed bag provided with the pulk. Separation distances between the antenna pairs were 0.09 m, 0.32 m, 0.66 m, and 1.07 m. Traces were sampled at a frequency of 42 GHz, whereby individual traces were recorded every 5 cm along the transects. We used a MALA odometer to achieve the required high relative positioning accuracy. Note that this setup was optimized to allow for accurate measurements in rather shallow snowpack. Deeper snowpack may require lower antenna frequencies, larger antenna offsets,

and more antenna pairings.

To test the capability of the GPR system in mountainous terrain (Fig. 3), measurements along ten transects were carried out. Overpasses of exactly the same transects with the GPR were repeated several times during snowmelt periods without precipitation in between the measurements. The dates of the measurements are listed in Table 1.

### 2.2.2   Manual measurements

Snow water equivalent was measured with a standard Federal sampler and thus only imposed minimal disturbance of the snowpack. In some cases however, when ice layers were present in the snowpack, taking the measurements became more challenging and several attempts were necessary to extract an intact snow core. For each snow water equivalent sample, three snow depth measurements were taken at the same location as well as one and two meters away along the transect.

Measurements of the liquid water content were taken using a Denoth meter (Denoth, 1994) and a small snow sampler for
concurrent snow density measurements. This required a snow pit which was dug sideways towards the center of the transect to minimize disturbances. We limited these liquid water content measurements to one vertical profile per transect and acquisition date. Complementary measurements some meters away from the transect, however, showed liquid water content to be fairly constant along individual transects. We consequently assumed liquid water content to be constant along individual transects.

### 2.3   Data Post Processing

To improve the signal-to-noise ratio and the visibility of reflecting layers, especially of the soil surface, several processing steps were applied to the GPR radargram before picking the relevant layer interfaces. First, a DC-shift was applied. This is a filter that removes an existing constant offset on each trace. Second, a gain filter was applied to amplify the signal as it attenuates within increasing travel time. A Kirchhoff migration was further applied to all radargrams of the S3 transect in Sertig valley as this site featured some roughness elements on the ground along the transect. All above steps were performed using the

Software ReflexW, Sandmeier Scientific Software. This software was further used to pick reflections of the layer interfaces using the phase follower utility built into ReflexW; this to determine the travel time of the direct wave, which is the first radar signal reaching the receiver, and of the bottom wave, which reaches the receiver after being reflected at the bottom interface underneath the snowpack. Then, odometer data were used to shift the picks of the individual antenna pairings to refer to a





common mid-point (CMP). This as well as the subsequent CMP analysis were performed in Matlab.

Procedures how to analyze CMP data are described in detail e.g. in Gustafsson et al. (2012). However, some processing details specific to our set up are detailed below. Since the antennas were very close above the snow surface at all times, the direct wave and the surface wave are in this work assumed to be identical. This implies that the direct wave velocity cannot be assumed

to be $0.3\,\mathrm{m\,ns^{-1}}$ as usual, but instead depends on material properties of the base of the sled (baseboard, air, bag, plastic shell), as well as of the top snow surface. While the former are constant, the latter are not. The direct wave velocity is needed to determine the two-way travel time from the picks. Even if the two-way travel time is insensitive to potential errors in the direct wave velocity due to variable effects from the snow surface, we have adopted the habit to determine the direct wave velocity for every GPR transect individually by optimizing the fit to complementary snow depth and snow water equivalent data collected

along each transect. The liquid water content could either be set to the manual measurements available for each individual transect and acquisition date, or included as a calibration parameter in the above optimization. In this work, we decided on the latter approach. Dielectric properties were estimated based on Tiuri et al. (1984).

## 3  Results

At all transects, snow water equivalent decreased between any series of consecutive overpasses. Subsequent measurements of

snow water equivalent profiles show mostly similar spatial patterns, which means that local minima in snow depth and snow water equivalent often remained in place during the entire melting period.

Fig. 4 (left panel) presents GPR-based snow depth estimates along transect S3 in the Sertig valley at all sampling dates. The GPR data is in very good agreement with the manual snow measurements (corresponding dots). The root-mean-square error (RMSE) evaluated over all reference points is 2.2 cm for snow depth and 22 mm for snow water equivalent. These values are on

the order of the accuracy at which these properties can be determined with manual measurements. Calculating depletion rates by subtracting subsequent acquisitions could also be tested against observations, as the manual measurements were conducted at fix positions. For these differential measurements of snow depth and of snow water equivalent the RMSE is 2.1 cm and 27 mm, respectively.

Additionally processing the S3 radargrams using Kirchhoff migration did reveal some additional fine-structured details in the

resulting profiles (Fig. 4, right panel). Note, however, that these details were only partly maintained over consecutive acquisitions, which questions whether the migration actually corrected for uneven subsurface features below the snow, just introduced additional noise, or both.

Regardless, corresponding RMSE were mostly unchanged with values of 2.3 cm for snow depth, 19 mm for snow water equivalent, 2.6 cm for differential snow depth, and 30 mm for differential snow water equivalent. These findings led us to abandon

migration as part of the standard post-processing steps.

Further examples of consecutive GPR acquisitions are presented in Fig. 5, however these are for the resulting snow water equivalent profiles along transects S2 and M1. Similar to the snow depth profiles in Fig. 4, but more prominently seen in the snow water equivalent profiles presented in Fig. 5, there are some deviating features from the common signature along the





profile, e.g. in S2 (Fig. 5, left panel) for the acquisition on 16 April 2015 around the 15-m mark. These deviations might simply stem from wrong interpretations within individual radargrams, but they might also result from a short deviation from the exact trajectory of the transect.

Validation data for all transects are presented in Table 2 and visualized in Fig. 6. The RMSE for snow depth using all tran-
sects in both valleys is 4.5 cm and 25 mm for snow water equivalent, 4.6 cm for differential snow depth (Fig. 6, top left), and 36 mm for differential snow water equivalent (Fig. 6, top right). These values were considerably augmented due to transect S1. This particular transect covered some very steep sections in which it was difficult to control the sled in order to follow the given trajectory. Misalignment between consecutive GPR acquisitions must have caused a considerably worse validation performance compared to all other transects. Removing S1 from the summary statistics results in vastly improved performance
(Fig. 6 bottom panels). Overall, we note that differential snow depth and differential snow water equivalent could be measured to a RMSE as low as 4 cm and 26 mm, respectively, provided that the experimental conditions allowed exact alignment of repeated GPR acquisitions.

Further examination of individual outliers revealed another source of experimental error. Manually measured and GPR-based data of S3 were found to be in excellent agreement with the exception of one individual point, encircled in Fig. 7. In this
example, manual observations of differential snow depth were significantly lower than corresponding GPR estimates. Fig. 7 (right panel) suggests that it is in fact the manual observation that is questionable, not the GPR data, and it appears as if the last reading of snow depth on 24 April 2015 at the third marker point was in error, possibly due to an ice layer. In assessing the accuracy of GPR systems, errors in the validation data therefore also have to be considered, and GPR systems may in fact produce more reliable snow depth estimates compared to traditional probing in certain instances.

## 4   Discussion

Our results show that the GPR system tested in this work was capable of measuring both snow depth and snow water equiv-alent in very good agreement with concurrent manual measurements. Direct comparison between GPR-based and manual observations resulted in an overall RMSE of 4.5 cm and 25 mm for snow depth and snow water equivalent, respectively. Any mismatches could have resulted from errors in the GPR-based estimates, in the manual observations, in the spatial misalign-
ment between measurement locations, or in all of these. Not only GPR-based estimates can be inaccurate. Probing errors do occur, e.g. if an ice layer is hit instead of the ground (Fig. 7), or if soft ground is penetrated. Misalignment errors were reported for profile S1, which resulted in considerably worse RMSE values (Table 2). Considering all sources of mismatch, we may infer that the accuracy of the GPR-based estimates of snow depth and snow water equivalent is not substantially different from those of manual measurements taken in a field survey setting. A similar GPR setup has been tested by Bühler et al. (2015),
but over a much more extended range of snow depths ranging from 0.7 to 2.7 m. They also found GPR and concurrent manual snow depth estimates to match very well, i.e. with an $R^2$ of 0.96 and RMSE of 7 cm.

The above findings are particularly noteworthy given that all campaigns have been done during the snow melt period where liquid water was present in the snowpack. Many previous applications of GPR have reported difficulties or reduced accuracy





under melting snow conditions. In our analysis, the liquid water content was assumed to be constant along a given transect. For the relatively short profiles investigated here (48 to 206 m), this simplified approach seemed to be feasible. However, for the use of GPR along longer transects, the lines should be broken into shorter segments of constant liquid water content. In this case, a set of roughly twelve snow depth and four snow water equivalent manual measurements is suggested to complement

each of those segments for calibration and validation purposes.

Note that in the two investigated areas the underlying ground surface was mainly alpine meadows. For this case, we assumed that a few post processing steps were sufficient to analyze the radargrams. Migration technique did not significantly improve the results, but might become necessary over more complex terrain.

Compared to manual measurements, the mobile CMP setup used in this study was able to (a) record snow depth and snow water

equivalent simultaneously, and (b) perform substantially faster and more efficient, even if the time for data post processing is included. Essentially, it can record data along transects at a spatial resolution that may be practically unavailable with manual measurement techniques. Therefore, such a GPR system should have a great potential when used in tandem with 3D airborne lidar surveys. Even if GPR-based acquisitions of snow density and snow water equivalent cannot provide full 3D-coverage over larger areas, a set of transects representing the physiographic range of the surveyed area would allow for a local, detailed,

and observation-based density parameterization to accurately convert lidar-based snow depth maps into corresponding snow water equivalent maps.

Interestingly and novel to our knowledge, RMSE values for differential acquisitions of snow ablation rates were similar to those determined for respective absolute acquisitions of snow depth and snow water equivalent. We can therefore infer that GPR systems can in fact be used to measure snow ablation rates in alpine terrain. For differential acquisitions, however, the

accurate spatial alignment of repeated overpasses is a key to arrive at accurate ablation rates. The practicalities to achieve this are challenging and may not be suited for operational or large-scale applications. Reducing the weight and size of the GPR system by implementing a shifted CMP approach certainly helped to maneuver the sled through alpine terrain, even under challenging conditions such as steep slopes (Fig. 3).

## 5   Conclusions

GPR surveys with mobile setups enable measuring snow depth, snow water equivalent, and snow density simultaneously. This technology is therefore particularly valuable for research-oriented or application-driven measurements of snow water resources. The setup presented here was optimized for efficient measurements in alpine terrain. To this end, a CMP assembly with four antenna offsets was mounted on a plastic sled, which was small enough to permit safe and convenient operations. Weight could be saved considerably by adopting an antenna setup referred to as shifted CMP design. This allowed us to

take continuous measurements of the above snow properties along line transects at a speed of roughly 25 meters per minute (excluding concurrent manual measurements).

Continuous CMP profiling allowed to arrive at snow depth, water equivalent, and density estimates without requiring bulk assumptions about the relationship between these properties. We have shown that these GPR measurements can be as accurate





as equivalent manual measurements in a field survey setting, in particular if the ground surface below the snow represent a good reflector and is considerably smooth. While extended datasets of accurate and collocated measurements of snow depth and snow water equivalent are valuable in itself, they have an even greater potential if acquired in parallel with 3D airborne lidar surveys. GPR-based measurements cannot provide full 3D-coverage over extended areas, but GPR-based data from sites which represent the physiographic range of the surveyed area would allow for an accurate conversion of lidar-based snow depth maps into equivalent snow water equivalent maps.

Further field tests demonstrated that the GPR system was even capable of measuring snow ablation rates. To our knowledge, this is the first study to demonstrate an appropriate non-destructive mobile setup which is agile and light enough to deliver such accurate differential values of snow properties. For that purpose, acquisitions were conducted repeatedly along the same transects over the course of several days to weeks. The accuracy of such differential measurements was similar to that of absolute measurements. However, the accurate spatial alignment of repeated overpasses was found to be a key factor in order to arrive at accurate ablation rates. The practicalities to achieve this were challenging and potentially not yet suited for operational or large-sale applications, but certainly feasible for research purposes.

*Acknowledgements.* Substantial field support was provided by Pascal Egli, Timea Marekova, and Giulia Mazzotti from the Snow Hydrology Group of the WSL Institute for Snow and Avalanche Research SLF. The authors thank Nathalie Chardon for reviewing the English of this article.



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



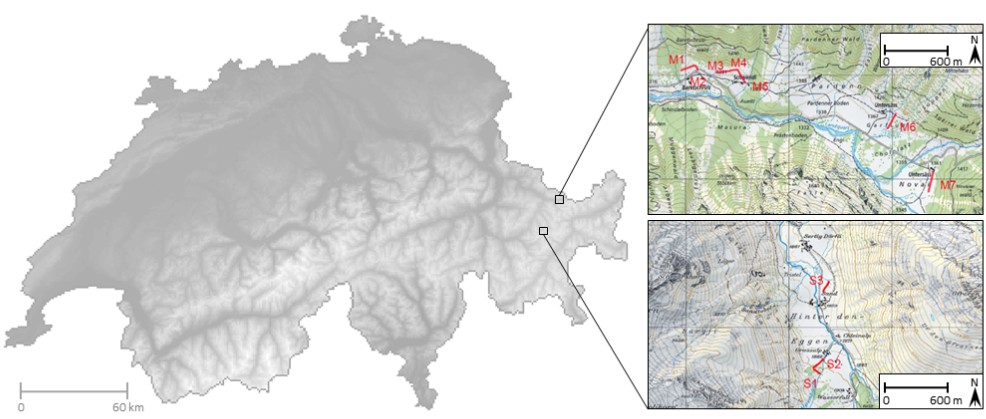

**Figure 1.** Topographic maps of Monbiel (top right) and Sertig (bottom right) valleys with sites indicated in red. Both valleys are located in the eastern Swiss Alps, the Monbiel valley being east-west orientated and 550 m lower than the south-north orientated Sertig valley. Reproduced by permission of swisstopo (JA100118). pixmaps©2016 swisstopo (5704 000 000).

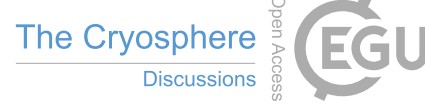

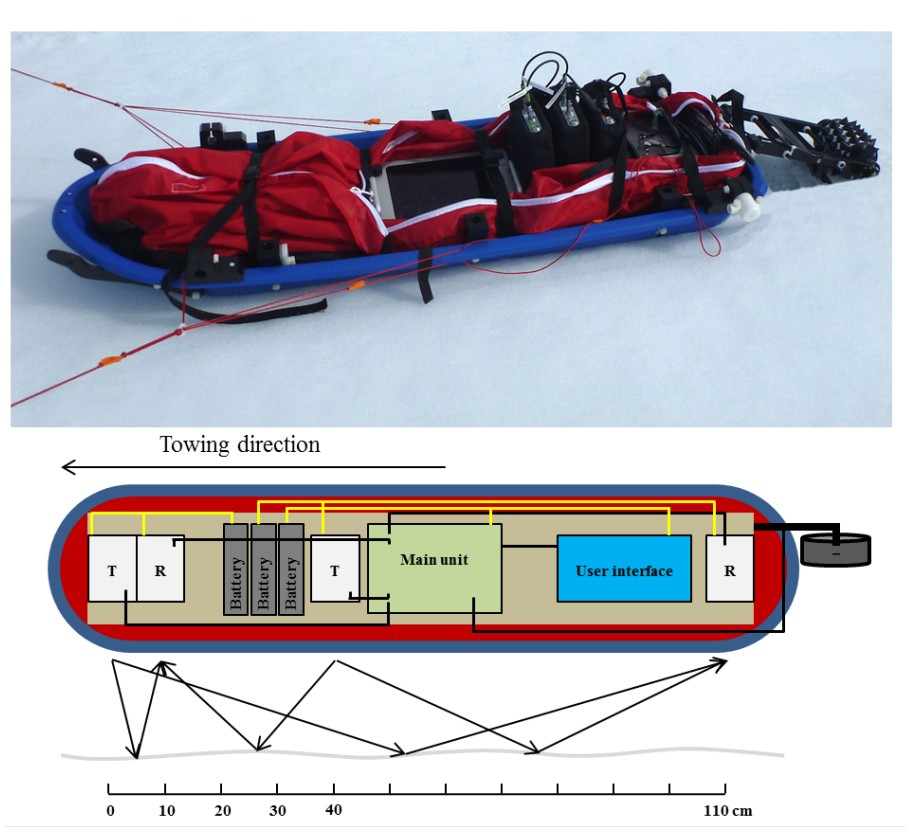

**Figure 2.** Setup illustration of the shifted CMP approach as topview (bottom) and picture taken in the field (top). T and R denote transmitting and receiving antennas on the sled, where arrows indicate the midpoint of each individual antenna pairing on a reflecting target. The odometer (grey wheel) is placed behind the sled. Antennas, main unit, user interface, batteries, and cables are placed on the sled's baseboard and enveloped in a waterproofed bag (red) inside the plastic sled (blue).





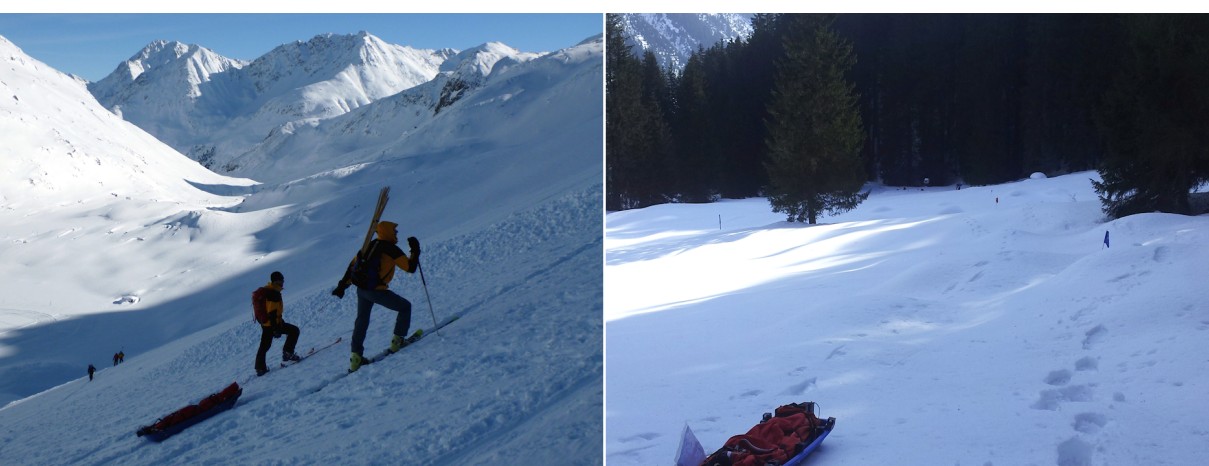

**Figure 3.** Fieldwork using GPR in steep alpine terrain (left) and (right) fieldwork at the third overpass of a transect in Monbiel valley. Flags mark points of manual measurements of snow depth and snow water equivalent for validation/calibration purposes, and serve as reference points for the synchronization of subsequent overpasses.





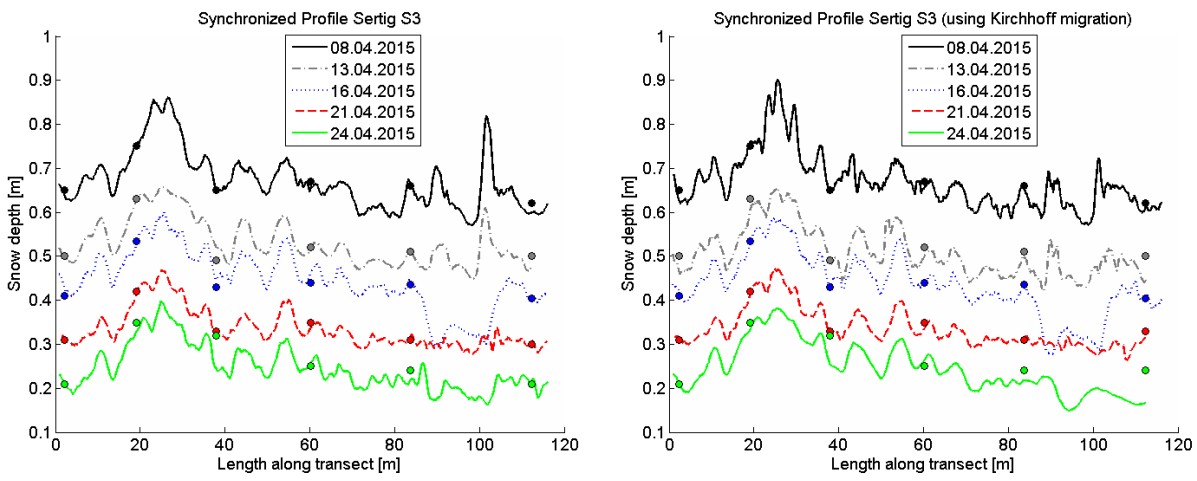

**Figure 4.** Snow depth of five overpasses along transect S3 without using migration (left panel) and with using Kirchhoff migration (right panel).





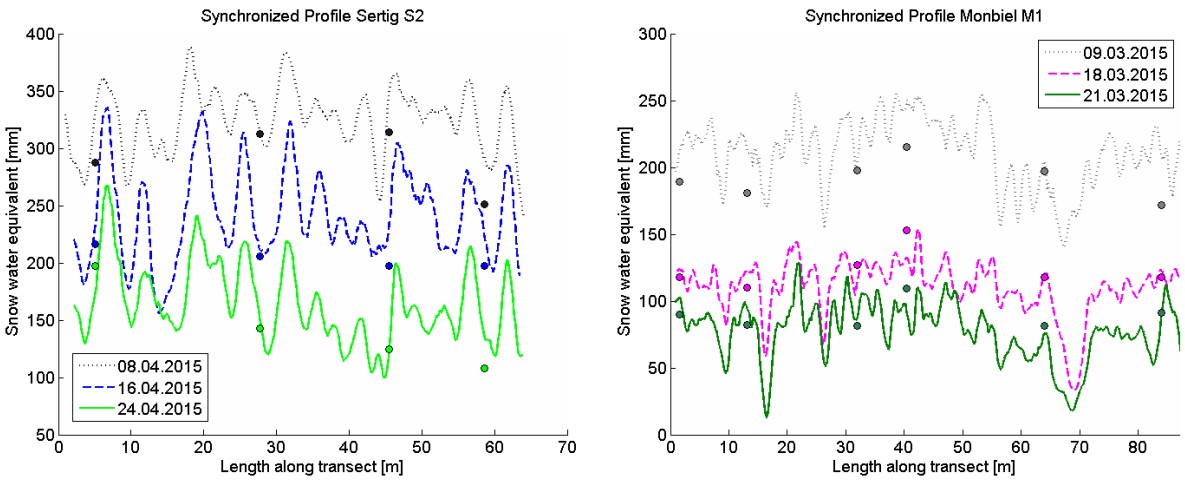

**Figure 5.** Snow water equivalent of three overpasses along transect S2 (left panel) and transect M1 (right panel).





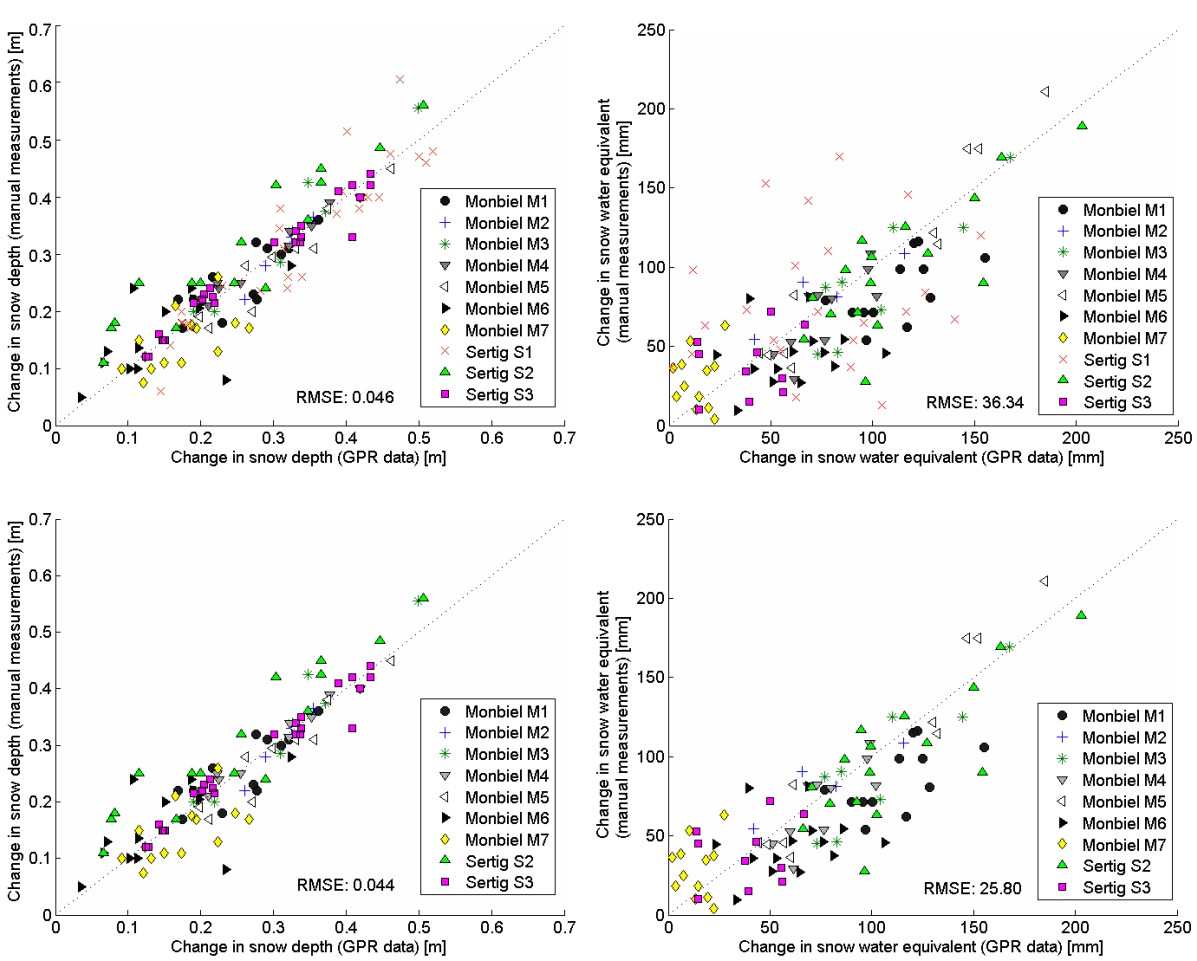

**Figure 6.** Relation between radar and manual measurements of differential snow depth (top left) and differential snow water equivalent (top right) using all transects, and differential snow depth (bottom left) and differential snow water equivalent (bottom right) using all transects except for S1.





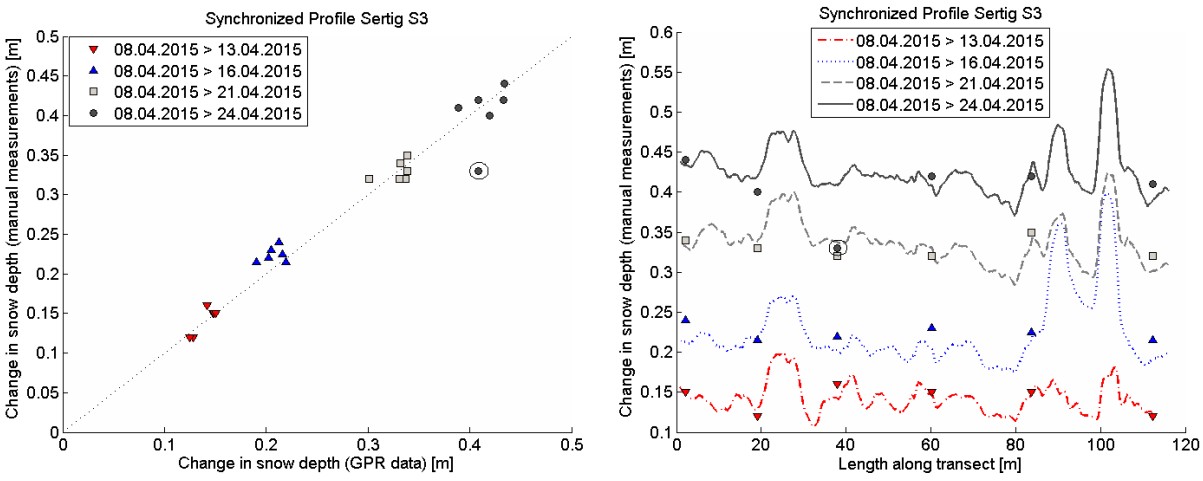

**Figure 7.** Relation between radar and manual measurements of differential snow depth at transect S3 (left panel), where an obvious error (encircled) could be attributed to a probing error during the last acquisition on 24 April 2015 (right panel).



**Table 1.** Characteristics of transects in this study. TM1: 09 March 2015, TM2: 11 March 2015, TM3: 18 March 2015, TM4: 21 March 2015, TS1: 08 April 2015, TS2: 13 April 2015, TS3: 16 April 2015, TS4: 21 April 2015, TS5: 24 April 2015

| Site name | Valley | Mean elevation [m a.s.l.] | Length [m] | Dates of measurements |
|---|---|---|---|---|
| M1 | Monbiel | 1342 | 87 | TM1, TM3, TM4 |
| M2 | Monbiel | 1337 | 48 | TM1, TM3, TM4 |
| M3 | Monbiel | 1359 | 206 | TM1, TM3, TM4 |
| M4 | Monbiel | 1363 | 133 | TM1, TM3, TM4 |
| M5 | Monbiel | 1374 | 103 | TM1, TM3, TM4 |
| M6 | Monbiel | 1361 | 165 | TM2, TM3, TM4 |
| M7 | Monbiel | 1354 | 174 | TM2, TM3, TM4 |
| S1 | Sertig | 1893 | 137 | TS1, TS2, TS3, TS4, TS5 |
| S2 | Sertig | 1903 | 63 | TS1, TS2, TS3, TS4, TS5 |
| S3 | Sertig | 1857 | 118 | TS1, TS2, TS3, TS4, TS5 |





**Table 2.** Root-mean-square error (RMSE) of snow depth, differential snow depth, snow water equivalent, and differential snow water equivalent for each transect.

| Site name | RMSE snow depth [cm] | RMSE differential snow depth [cm] | RMSE snow water equivalent [mm] | RMSE differential snow water equivalent [mm] |
|-----------|----------------------|-----------------------------------|---------------------------------|----------------------------------------------|
| M1 | 3.7 | 3.5 | 17.2 | 32.0 |
| M2 | 2.5 | 2.1 | 8.2 | 14.3 |
| M3 | 2.9 | 3.6 | 13.2 | 21.9 |
| M4 | 2.9 | 1.6 | 17.4 | 15.6 |
| M5 | 5.1 | 3.1 | 21.7 | 18.6 |
| M6 | 4.9 | 6.3 | 17.6 | 30.5 |
| M7 | 5.6 | 5.1 | 19.6 | 26.1 |
| S1 | 5.0 | 5.1 | 39.8 | 60.5 |
| S2 | 6.9 | 7.0 | 29.6 | 28.1 |
| S3 | 2.2 | 2.1 | 21.9 | 26.6 |