# Peer review of "On measuring snow ablation rates in alpine terrain with a mobile GPR device"

_The Cryosphere, 2016_

## Referee Comment (RC1) · Anonymous Referee #1 · 14 Apr 2017

Dear Editor, I have read carefully the paper from Griessinger et al., untitled "On measuring snow ablation rates in alpine terrain with a mobile GPR device", overall the paper is well written and easily understandable. The topic is very interesting and definitely have a strong interest for the scientific community. However some more precision needs to be added to this work, especially in term of GPR radargram quality, and snow density retrieved from GPR measurement. I would consider this paper for publication after major revision. I will be happy to re-read the corrected version.

General comments: Very interested study, very good experimental setup, very ingenious way to take measurement without affecting the snow compaction. My only concerns are related to your radargram quality and your way of retrieving the snow density/snow water equivalent from GPR:

[Figure]

- I think it would greatly ameliorate the paper by adding a radargram, and the picking of your reflection of interest.

- I think it would greatly ameliorate the paper by explaining a bit more, how you did calibrated you GPR to infer SWE and Snow density, the paper of Gustafsson is a good example.

Other comments:

Page 2 Line 5: "Many applications of GPR to measure spatially distributed snow properties are generally conducted when dry snow conditions are present" Would that be helpful to the reader to define a bit dry snow conditions?

"The sled was towed by two persons, one to the left and one to the right, so that it was not required to step on the transects." Very good

Page 4 line 4 "This way, the antennas were placed approximately 2 cm above the snow surface" Taking into account during the processing?

Page 4 line 7 "Traces were sampled at a frequency of 42 GHz, whereby individual traces were recorded every 5 cm along the transects." It would be helpful to know what was your time windows to have as well you time sampling, since you are looking at very fine velocity variations.

Page 4 line 8 "We used a MALA odometer to achieve the required high relative positioning accuracy" The wheel you have in your picture is made from Mala or handmade? Did you re-calibrate the odometer before each of your survey? Since the snow conditions can be different, the slippering of the wheel can be different.

Page 4 line 12 "Overpasses of exactly the same transects with the GPR were repeated several times during snowmelt periods without precipitation in between the measurements" From what I understand, you are passing on the same transect every time, what could you say about the impact of your repeated transect on the snow density?

Page 4 line 21 " This required a snow pit which was dug sideways towards the center of the transect to minimize disturbances" Did you make a new one every-time you surveyed?

Page 4 line 25 " First, a DC-shift was applied. This is a filter that removes an existing constant offset on each trace. Second, a gain filter was applied to amplify the signal as it attenuates within increasing travel time." Maybe would it be simpler to just say that you removed the lower frequency from the data, or "de-wowed" them? Could you precise what was you gain applied? AGC, exponential ? And once again as already mentionned would be nice to see radargrams, before and after processing, and your picked reflection too?

Page 4 line 27, " A Kirchhoff migration was further applied to all radargrams " You determined the velocity by the direct wave? from which Tx and Rx, the long or short spacing?

Page 5 Line 8, "Even if the two-way travel time is insensitive to potential errors in the direct wave velocity due to variable effects from the snow surface" Please could you rephrase, you are saying that it as no effect but since you are making a TWT calculation based on the direct wave it as effect, maybe you are implying that the impact of the near surface snow has no effect on the direct wave Travel Time, in this case, could you re-phrase in agreement with Line 5 and 6 of the same page?

Page 5 line 12, "Dielectric properties were estimated based on Tiuri et al. (1984)." Could you give a little bit more information on why you decided to use the model of Tiuri et al. ? In addition I think the paper is missing as little more complete explanation on the way you retrieved the SWE. I guess you used what was done in Gustavson et al., however the paper is missing your calibration parameter for the snow density (In Gustavson, page 4).

Figure 4 -Can you please edits what are the points in the caption of the Figure.

[Figure]

---

## Short Comment (SC1) · 10 May 2017

This is a well designed study that provides valuable information concerning the error associated with GPR measurements of snow. I have also recently published a paper using GPR to measure ablation rates in the subalpine that may be of interest to you for this study (Webb, 2017; doi: 10.1007/s11707-017-0645-0)

I agree with the reviewer that showing a radargram would be beneficial, as well as further detailing your methods.

One addition that I think would greatly add to the article is estimating how total SWE or ablation values are improved along an entire transect through the use of GPR. Comparing the classic pit and depth probe measurements to GPR surveys for total SWE along each transect. This would add a lot of additional value to the paper. It looks like

you began doing this with comparing depth measurements, but further discussion and clarification would be helpful.

Good work!

---

## Referee Comment (RC2) · Anonymous Referee #2 · 22 Jun 2017

The paper by Griessinger et al presents spatially-distributed measurements along transects (of typically 100 m in length) of both snow depth and snow water equivalent (SWE), and their variation over time, considering periods from a few days to a couple of weeks. Those measurements were done with the help of a new mobile ground penetrating radar (GPR) device. The paper starts with a brief introduction on GPR technique overall and its application to snow hydrology. Section 2 describes the methods used by the authors: the two study areas considered, the measurements with the mobile GPR device (including the manual measurements for calibration and validation) and data post processing. Section 3 presents the results on snow depth and water equivalent measurements, and those results are discussed in section 4. Finally, section 5 concludes the paper.

General comments

It seems to me that the design of the mobile GPR device and its performance shown in the present paper are of interest for the snow hydrological research community. As such, the topic addressed in the paper could be worthy of publication to The Cryosphere journal. However, I believe that the manuscript in its current form is far from being suitable for publication because it suffers from a lack of detailed information on a number of crucial points regarding the methods, and there are some confusing parts additionally. I found the paper much too short. I have tried to summarize my concerns in the list of the specific comments provided below. I strongly encourage the authors to revise and extend a number of parts of their paper in order to make it self-contained, clearer and more convincing.

Specific comments

Abstract:

1) line 7: "over time" is seemingly too vague... please mention the typical time scales (from a few days to a couple of weeks) which you considered in your field study. Also, it would be nice to add in the abstract an information on the typical length of the profiles considered.

Section 1:

2) page 2, lines 20-23: I found the discussion on the techniques used to measure snow properties, in particular density, and SWE, much too short. Please could you extend it and explain how density and SWE are measured in your study?

Section 2.2:

3) page 3, line 15: please could you include example of radargrams and illustrate how your reference points are efficient for synchronization?

4) page 3, line 16: while mentioning the manual measurements here, you could refer

to section 2.2.2. Otherwise the reader may expect those manual measurements to be detailed here.

5) page 3, line 20: this appears to be a limitation of your mobile GPR device. Would you have any suggestion on a way of overcoming this in the future? Could you please make a comment (maybe you could include this point in conclusion too). Furthermore, I wonder how this "temporary" crust is thick and may affect the measurements, particularly concerning the assumptions made for estimates of density and SWE.

Section 2.2.2:

6) page 4, lines 22-23: please could you show this in a table (or plot)? What was the typical standard deviation?

Section 2.3:

7) page 4: the section is much too short. I invite you to go into much more detail on your method for data post-processing. In particular, I strongly suggest you to include radargrams corresponding to each of the main steps.

8) page 5, lines 2-10: this part needs strong revision. I understand here that you need your manual measurements of snow depth, density and SWE for calibration/optimization. Please explain better the optimization procedure. Could you show for instance a cross-comparison between the results with a direct wave velocity of 0.3 m ns-1 and the results with your optimization?

9) page 5, lines 10-12: this part is again too short and is confusing me. If I am not wrong I understand here that you finally chose to use an optimization procedure but not to consider manual measurements of SWE as a direct input to your GPR measurements. However, in the previous lines (see comment 8 above), you are saying that you used those manual measurements for fits. Please explain better your calibration/optimization procedure. In particular, it is not clear to me if there are some data points from 'manual measurements' displayed on Figs. 4, 5, 6 or 7 which were used for

calibration/optimization. If yes, you should label them explicitly. In the absence of much more detail on this, the frontier between calibration and validation concerning SWE is seemingly tiny.

10) page 5, last sentence: please could you give detail about the estimation proposed by Tiuri et al. (1984) to make your paper self-contained.

Section 3:

11) page 5, lines 24-25: please could you show the radargrams! It is quite frustrating not to see any radargram in your paper.

12) page 6, lines 9-10: I am not sure you need to repeat the two plots without S1. The snow depth on S1 works well but not the SWE. Please could you comment on that?

Section 4:

13) Page 7, lines 3-5: the calibration/optimization procedure for density and SWE needs much more explanation (see my comments 8 and 9 again).

Section 5:

14) Page 8: the last paragraph of that section appears to me as a reiteration of the last paragraph of section 4. I would suggest you to remove this paragraph from one of the two sections.

Technical corrections, typing errors, etc.

Abstract, line 10 (second-last sentence): I would suggest you to either remove the content between brackets or keep it but not between brackets.

section 1, page 2, lines 26, 27 and 29: in line 27 you are defining the term "snow ablation rates" for "change rates" but in line 26 (just above) you already use "snow ablation rates" AND in line 29 just below you use another term "snowmelt rates"... please fix this.

section 2.2.1, page 3, line 28: there is a typo ('parings')

Figures:

Fig. 1: could you please increase the size of the top and bottom right pictures? and try to improve them overall? I must say that they are not so easy to read.

Fig. 3: would be nice to indicate in the right photo -with an arrow for instance- the typical distance between two flags.

Fig. 4, 5 and 7: could you please add typical error bars for both the curves showing the continuous GPR measurements, and the data points for the discrete manual measurements?

Fig. 6: I suggest you to remove the two bottom plots without S1. I believe it is sufficient to keep the two top plots and indicate briefly in the caption why S1 is an outlier for SWE, and refer to text for explanation.

---

## Author Comment (AC1) · 28 Jul 2017

We would like to thank the reviewer for the positive feedback and the useful comments. Please find below our replies as inserted blue text.

Kind regards,

Nena Griessinger, Franziska Mohr and Tobias Jonas

Dear Editor, I have read carefully the paper from Griessinger et al., untitled "On measuring snow ablation rates in alpine terrain with a mobile GPR device", overall the paper is well written and easily understandable. The topic is very interesting and definitely have a strong interest for the scientific community. However some more precision needs to be added to this work, especially in term of GPR radargram quality, and snow density retrieved from GPR measurement. I would consider this paper for publication after major revision. I will be happy to re-read the corrected version.

General comments: Very interested study, very good experimental setup, very ingenious way to take measurement without affecting the snow compaction. My only concerns are related to your radargram quality and your way of retrieving the snow density/snow water equivalent from GPR:

- I think it would greatly ameliorate the paper by adding a radargram, and the picking of your reflection of interest.

Thank you for this suggestion, we will include a radargram with corresponding picks.

- I think it would greatly ameliorate the paper by explaining a bit more, how you did calibrated you GPR to infer SWE and Snow density, the paper of Gustafsson is a good example.

We will give more details on the calibration procedure in the methods section.

Other comments:

Page 2 Line 5: "Many applications of GPR to measure spatially distributed snow properties are generally conducted when dry snow conditions are present" Would that be helpful to the reader to define a bit dry snow conditions?

Bradford et al. (2009) give a classification of the wetness of snow. According to them, we define dry snow to have a volumetric water content of 0 %. We will clarify this in the manuscript.

"The sled was towed by two persons, one to the left and one to the right, so that it was not required to step on the transects." Very good

Thank you.

Page 4 line 4 "This way, the antennas were placed approximately 2 cm above the snow surface" Taking into account during the processing?

Yes, this offset was taken into account during the processing. We will add this information.

Page 4 line 7 "Traces were sampled at a frequency of 42 GHz, whereby individual traces were recorded every 5 cm along the transects." It would be helpful to know what was your time windows to have as well you time sampling, since you are looking at very fine velocity variations.

Recording was triggered using an odometer, so did not happen at fixed time intervals. Trace lengths were set to 744 samples per trace.

Page 4 line 8 "We used a MALA odometer to achieve the required high relative positioning accuracy" The wheel you have in your picture is made from Mala or handmade? Did you re-calibrate the odometer before each of your survey? Since the snow conditions can be different, the slippering of the wheel can be different.

Mala provides different types of odometers. For highest positioning accuracy, we used a hip chain odometer which measures the distance from a specialized thread that unwinds from a spool as the sled moves along the transect. This approach is independent of snow surface conditions and does not require recalibration before every survey. As a hip chain would not be visible in the image, we chose to present an image with a regular (custom-made) odometer wheel for illustration purposes. We will add this information to the manuscript.

Page 4 line 12 "Overpasses of exactly the same transects with the GPR were repeated several times during snowmelt periods without precipitation in between the measurements" From what I understand, you are passing on the same transect every time, what could you say about the impact of your repeated transect on the snow density?

Our fieldwork was carried out after clear, cold nights which guaranteed a crust on the snow surface (see Chapter 2.2). We are thus confident that we did not alter the snow density by the repeated measurements.

Page 4 line 21 " This required a snow pit which was dug sideways towards the center of the transect to minimize disturbances" Did you make a new one every-time you surveyed?

Yes, at every transect we dug a new snow pit for each survey because we expected the ablation rates to differ between a closed snow pack and a persisting snow pit due to radiation effects. We will clarify this.

Page 4 line 25 " First, a DC-shift was applied. This is a filter that removes an existing constant offset on each trace. Second, a gain filter was applied to amplify the signal as it attenuates within increasing travel time." Maybe would it be simpler to just say that you removed the lower frequency from the data, or "de-wowed"

them? Could you precise what was you gain applied? AGC, exponential ? And once again as already mentionned would be nice to see radargrams, before and after processing, and your picked reflection too?

We applied a manual y-gain filter where needed. The radargrams mostly showed a very clear signal so excessive filtering was not required. We will include an exemplary radargram to demonstrate this.

Page 4 line 27, " A Kirchhoff migration was further applied to all radargrams " You determined the velocity by the direct wave? from which Tx and Rx, the long or short spacing?

We used the snow pit data associated with the S3 transect to arrive at a radar velocity needed for the conversion within the migration procedure.

Page 5 Line 8, "Even if the two-way travel time is insensitive to potential errors in the direct wave velocity due to variable effects from the snow surface" Please could you rephrase, you are saying that it as no effect but since you are making a TWT calculation based on the direct wave it as effect, maybe you are implying that the impact of the near surface snow has no effect on the direct wave Travel Time, in this case, could you re-phrase in agreement with Line 5 and 6 of the same page?

Thank you, we will clarify the description.

Page 5 line 12, "Dielectric properties were estimated based on Tiuri et al. (1984)." Could you give a little bit more information on why you decided to use the model of Tiuri et al. ? In addition I think the paper is missing as little more complete explanation on the way you retrieved the SWE. I guess you used what was done in Gustavson et al., however the paper is missing your calibration parameter for the snow density (In Gustavson, page 4).

We compared the methods suggested by Frolov (Frolov at al., 1999), Tiuri (Tiuri et al., 1984) and Looyenga (Looyenga, 1965) and did not find considerable differences on the overall results. We used the parameterizations from Tiuri based on extended field tests at the Weissfluhjoch snow monitoring site (Davos Switzerland) by Koch et al. (2014). We agree that Section 2.3 should be extended, and we will provide more information in the updated manuscript.

Figure 4 -Can you please edits what are the points in the caption of the Figure.

The points display the position of the reference marker. Thank you, we will adapt the mentioned figure.

References:

Frolov, A. D. (1999). On dielectric properties of dry and wet snow. Hydrological processes, 13(12-13), 1755-1760.

Koch, F., Prasch, M., Schmid, L., Schweizer, J., & Mauser, W. (2014). Measuring snow liquid water content with low-cost GPS receivers. Sensors, 14(11), 20975-20999.

Looyenga, H. (1965). Dielectric constants of heterogeneous mixtures. Physica, 31(3), 401-406.

Tiuri, M., Sihvola, A., Nyfors, E., & Hallikaiken, M. (1984). The complex dielectric constant of snow at microwave frequencies. IEEE Journal of Oceanic Engineering, 9(5), 377-382.

---

## Author Comment (AC2) · 28 Jul 2017

Dear Ryan Webb,

we would like to thank you for the positive feedback and the useful comments.

Please find below our replies as inserted blue text.

Kind regards,

Nena Griessinger, Franziska Mohr and Tobias Jonas

This is a well designed study that provides valuable information concerning the error associated with GPR measurements of snow. I have also recently published a paper using GPR to measure ablation rates in the subalpine that may be of interest to you for this study (Webb, 2017; doi: 10.1007/s11707-017-0645-0)

Thank you for your positive feedback and the suggested literature which we will include.

I agree with the reviewer that showing a radargram would be beneficial, as well as further detailing your methods.

Yes, we will include a radargram and give more details on our methods.

One addition that I think would greatly add to the article is estimating how total SWE or ablation values are improved along an entire transect through the use of GPR. Comparing the classic pit and depth probe measurements to GPR surveys for total SWE along each transect. This would add a lot of additional value to the paper. It looks like you began doing this with comparing depth measurements, but further discussion and clarification would be helpful.

Unfortunately we have no additional snow data to comment on the improvements of GPR based SWE estimates over traditional manual measurements as this was not the purpose of our work.

---

## Author Comment (AC3) · 28 Jul 2017

We would like to thank the reviewer for the feedback and the useful comments. Please find below our replies as inserted blue text.

Kind regards,

Nena Griessinger, Franziska Mohr and Tobias Jonas

The paper by Griessinger et al presents spatially-distributed measurements along transects (of typically 100 m in length) of both snow depth and snow water equivalent (SWE), and their variation over time, considering periods from a few days to a couple of weeks. Those measurements were done with the help of a new mobile ground penetrating radar (GPR) device. The paper starts with a brief introduction on GPR technique overall and its application to snow hydrology. Section 2 describes the methods used by the authors: the two study areas considered, the measurements with the mobile GPR device (including the manual measurements for calibration and validation) and data post processing. Section 3 presents the results on snow depth and water equivalent measurements, and those results are discussed in section 4. Finally, section 5 concludes the paper.

General comments

It seems to me that the design of the mobile GPR device and its performance shown in the present paper are of interest for the snow hydrological research community. As such, the topic addressed in the paper could be worthy of publication to The Cryosphere journal. However, I believe that the manuscript in its current form is far from being suitable for publication because it suffers from a lack of detailed information on a number of crucial points regarding the methods, and there are some confusing parts additionally. I found the paper much too short. I have tried to summarize my concerns in the list of the specific comments provided below. I strongly encourage the authors to revise and extend a number of parts of their paper in order to make it self-contained, clearer and more convincing.

Specific comments

Abstract:

1) line 7: "over time" is seemingly too vague... please mention the typical time scales (from a few days to a couple of weeks) which you considered in your field study. Also, it would be nice to add in the abstract an information on the typical length of the profiles considered.

We will adapt the abstract accordingly.

Section 1:

2) page 2, lines 20-23: I found the discussion on the techniques used to measure snow properties, in particular density, and SWE, much too short. Please could you extend it and explain how density and SWE are measured in your study?

Thank you, we will extend the introduction. We will provide more information on our methods in Section 2 in order to keep the structure of the manuscript clear.

Section 2.2:

3) page 3, line 15: please could you include example of radargrams and illustrate how your reference points are efficient for synchronization?

Thank you, we will include a radargram. The reference points are fix positions marked in the field and recorded in the radargram. The positioning of traces within radargrams of repeated overpasses was forced to exactly match at these points. This way the positioning accuracy in between those points and consequently along the entire transect can be limited to below 5 centimeters. We will add this information in Section 2.3.

4) page 3, line 16: while mentioning the manual measurements here, you could refer to section 2.2.2. Otherwise the reader may expect those manual measurements to be detailed here.

Thank you, we will refer to Section 2.2.2.

5) page 3, line 20: this appears to be a limitation of your mobile GPR device. Would you have any suggestion on a way of overcoming this in the future? Could you please make a comment (maybe you could include this point in conclusion too). Furthermore, I wonder how this "temporary" crust is thick and may affect the measurements, particularly concerning the assumptions made for estimates of density and SWE.

Thank you for this comment. The mobile GPR device itself could also be used on any snow surface. However repeated surveys may only make sense if the snowpack remains as undisturbed as possible to allow meaningful differential measurements. Hence our strategy was to limit measurements to mornings after a clear night.

Section 2.2.2:

6) page 4, lines 22-23: please could you show this in a table (or plot)? What was the typical standard deviation?

Standard deviation (SD) between repeated LWC measurements within transects were 0.48 % VWC (mean SD), 0.75 % VWC (max SD).

Section 2.3:

7) page 4: the section is much too short. I invite you to go into much more detail on your method for data post-processing. In particular, I strongly suggest you to include radargrams corresponding to each of the main steps.

We agree and will extend this section.

8) page 5, lines 2-10: this part needs strong revision. I understand here that you need your manual measurements of snow depth, density and SWE for calibration/optimization. Please explain better the optimization procedure. Could you show for instance a cross-comparison between the results with a direct wave velocity of 0.3 m ns-1 and the results with your optimization?

We will give more details on the calibration procedure in the methods section.

9) page 5, lines 10-12: this part is again too short and is confusing me. If I am not wrong I understand here that you finally chose to use an optimization procedure but not to consider manual measurements of SWE as a direct input to your GPR measurements. However, in the previous lines (see comment 8 above), you are saying that you used those manual measurements for fits. Please explain better your calibration/optimization procedure. In particular, it is not clear to me if there are some data points from 'manual measurements' displayed on Figs. 4, 5, 6 or 7 which were used for calibration/optimization. If yes, you should label them explicitly. In the absence of much more detail on this, the frontier between calibration and validation concerning SWE is seemingly tiny.

We are sorry for the confusion. The points in figure 4, 5, 6, and 7 display the position of the reference markers which are collocated with locations of manual measurements. Added information on the calibration procedure will clarify why manual snow measurements are not used as direct input but yet involved as part of the processing.

10) page 5, last sentence: please could you give detail about the estimation proposed by Tiuri et al. (1984) to make your paper self-contained.

We will consider providing further details from Tiuri et al. (1984).

Section 3:

11) page 5, lines 24-25: please could you show the radargrams! It is quite frustrating not to see any radargram in your paper.

Yes, we will include a radargram.

12) page 6, lines 9-10: I am not sure you need to repeat the two plots without S1. The snow depth on S1 works well but not the SWE. Please could you comment on that?

We will consider this suggestion. However it seems interesting to us to demonstrate the effect of a single problematic transect on the overall results (and how they can differ between snow depth and snow water equivalent). Misalignment problems between repeated overpasses at S1 had a considerably impact on differential snow water equivalent measurements which was not the case for differential snow depth measurements. This is because snow depth measurement can and were repeatedly taken at the exact same position (marker points) which was not possible for (destructive) snow water equivalent measurements.

Section 4:

13) Page 7, lines 3-5: the calibration/optimization procedure for density and SWE needs much more explanation (see my comments 8 and 9 again).

See above.

Section 5:

14) Page 8: the last paragraph of that section appears to me as a reiteration of the last paragraph of section 4. I would suggest you to remove this paragraph from one of the two sections.

Thank you.

Technical corrections, typing errors, etc.
Abstract, line 10 (second-last sentence): I would suggest you to either remove the content between brackets or keep it but not between brackets.

We will consider your suggestion.

section 1, page 2, lines 26, 27 and 29: in line 27 you are defining the term "snow ablation rates" for "change rates" but in line 26 (just above) you already use "snow ablation rates" AND in line 29 just below you use another term "snowmelt rates"... please fix this.

OK.

section 2.2.1, page 3, line 28: there is a typo ('parings')

OK.

Figures:
Fig. 1: could you please increase the size of the top and bottom right pictures? and try to improve them overall? I must say that they are not so easy to read.

OK.

Fig. 3: would be nice to indicate in the right photo -with an arrow for instance- the typical distance between two flags.

Thank you for the suggestion.

Fig. 4, 5 and 7: could you please add typical error bars for both the curves showing the continuous GPR measurements, and the data points for the discrete manual measurements?

We will consider your suggestion; note however that there was only one manual snow measurement per reference point.

Fig. 6: I suggest you to remove the two bottom plots without S1. I believe it is sufficient to keep the two top plots and indicate briefly in the caption why S1 is an outlier for SWE, and refer to text for explanation.

Thank you.